# Exploring PLGA-OH-CATH30 Microspheres for Oral Therapy of *Escherichia coli*-Induced Enteritis

**DOI:** 10.3390/biom14010086

**Published:** 2024-01-10

**Authors:** Xiaoqian Jiao, Bin Liu, Xufeng Dong, Shubai Wang, Xiulei Cai, Hongliang Zhang, Zhihua Qin

**Affiliations:** College of Veterinary Medicine, Qingdao Agricultural University, Qingdao 266109, China; xqj426@stu.qau.edu.cn (X.J.); liubin@stu.qau.edu.cn (B.L.); dongxufeng@qau.edu.cn (X.D.); 199101020@qau.edu.cn (S.W.); caixiulei@qau.edu.cn (X.C.); 201302018@qau.edu.cn (H.Z.)

**Keywords:** *Escherichia coli*, enteritis, PLGA microspheres, antimicrobial peptide, inflammatory, gut microbiome

## Abstract

Antibiotic therapy effectively addresses *Escherichia coli*-induced enteric diseases, but its excessive utilization results in microbial imbalance and heightened resistance. This study evaluates the therapeutic efficacy of orally administered poly (lactic-co-glycolic acid) (PLGA)-loaded antimicrobial peptide OH-CATH30 microspheres in murine bacterial enteritis. Mice were categorized into the healthy control group (CG), untreated model group (MG), OH-CATH30 treatment group (OC), PLGA-OH-CATH30 treatment group (POC), and gentamicin sulfate treatment group (GS). Except for the control group, all other experimental groups underwent *Escherichia coli*-induced enteritis, followed by a 5-day treatment period. The evaluation encompassed clinical symptoms, intestinal morphology, blood parameters, inflammatory response, and gut microbiota. PLGA-OH-CATH30 microspheres significantly alleviated weight loss and intestinal damage while also reducing the infection-induced increase in spleen index. Furthermore, these microspheres normalized white blood cell count and neutrophil ratio, suppressed inflammatory factors (IL-1β, IL-6, and TNF-α), and elevated the anti-inflammatory factor IL-10. Analysis of 16S rRNA sequencing results demonstrated that microsphere treatment increased the abundance of beneficial bacteria, including *Phocaeicola vulgatus*, in the intestinal tract while concurrently decreasing the abundance of pathogenic bacteria, such as *Escherichia*. In conclusion, PLGA-OH-CATH30 microspheres have the potential to ameliorate intestinal damage and modulate the intestinal microbiota, making them a promising alternative to antibiotics for treating enteric diseases induced by *Escherichia coli*.

## 1. Introduction

*Escherichia coli* (*E. coli*), ubiquitous microorganisms in the intestinal tracts of humans and animals, play a crucial role in the gut microbiome [1]. However, an excess of *E. coli* can disturb the intricate balance of gut microbial communities, potentially making them pathogenic [2]. In this context, *E. coli* may precipitate a diverse array of diseases, among which ulcerative colitis (UC) serves as a notable exemplar [3]. UC entails multifaceted interplays involving gut microbiota, host genetics, immunological elements, and environmental variables [4,5]. The conventional therapeutic approach for addressing intestinal maladies attributable to *E. coli* has revolved around the administration of antibiotics. Nonetheless, *E. coli* manifests a myriad of serotypes and retains susceptibility to resistance development [6]. Prolonged or injudicious antibiotic employment may engender the dissemination of antibiotic-resistant genes within the gut microbiota of both human and animal hosts [7]. This phenomenon disrupts the balance of the intestinal microbiota, affecting the normal microbial ecosystem and compromising the functionality of the mucosal barrier. These consequences culminate in untoward effects, such as antibiotic-associated diarrhea (AAD) [8,9].

Furthermore, a recent report by the World Health Organization underscores the inadequacy of antibiotics currently under development in addressing infections and complications arising from multidrug resistance [10]. The pace of resistance escalation far outstrips that of antibiotic development [11]. Consequently, an imperative exists to innovate and develop novel antimicrobial agents to combat bacterial infections, given the relentless emergence of antibiotic resistance. Antimicrobial peptides (AMPs), a class of low-molecular-weight peptides endowed with broad-spectrum antimicrobial activity, have gained significant attention as potential alternatives to antibiotics due to their intrinsic resistance mitigation properties [12]. Nevertheless, the stability of antimicrobial peptides remains susceptible to environmental influences, thereby constraining their applicability in oral drug delivery systems [13]. To overcome this challenge, incorporating drug delivery systems shows promise in enhancing both the bioavailability and stability of antimicrobial peptides.

OH-CATH30, a naturally occurring linear cationic peptide in the king cobra, consists of 30 amino acids and falls within the antimicrobial peptide family [14]. It exhibits strong antimicrobial effects against various pathogens, with minimal cytotoxicity and hemolytic activity. The peptide’s potent antimicrobial properties, coupled with its ability to selectively modulate chemokines and cytokines, are pivotal in protecting animals from life-threatening sepsis [15,16]. Biodegradable organic compounds include poly (lactic-co-glycolic acid) (PLGA), a copolymer formed by the random polymerization of lactic acid (LA) and glycolic acid (GA) [17]. The hydrolysis product of PLGA, including LA and GA, actively participates in organism metabolic processes, ultimately producing carbon dioxide and water for excretion, showcasing commendable biocompatibility and degradability [18]. Functioning as a prevalent drug delivery vehicle with controlled-release capabilities, PLGA finds extensive use in biomedical research due to its exceptional performance. In our previous investigation, we developed PLGA-OH-CATH30 microspheres for the treatment of bacterial keratitis in companion animals. The microspheres exhibit a high encapsulation efficiency of 75.2 ± 3.62% and a loading capacity of 18.25 ± 5.73%. With a particle size distribution ranging from 200 to 1000 nm and a ζ potential of −17.3 ± 1.91 mV, the microspheres displayed uniform and smooth characteristics. Additionally, they demonstrated sustained release properties. Rigorous in vitro and in vivo tolerance assessments confirmed the excellent biocompatibility of the antimicrobial peptide-loaded microspheres, causing no significant irritation to ocular tissues. Notably, remarkable antibacterial efficacy was observed in both in vitro and in vivo experiments. Subsequent to encapsulation, these microspheres significantly enhanced the bioavailability and stability of the antimicrobial peptide, providing a direct and convenient application for therapeutic use [19]. 

Building on the success of microsphere technology in treating pet keratitis, we explored the potential of PLGA-OH-CATH30 microspheres to manage murine enteritis and assessed their efficacy. Utilizing microsphere technology, our objective is to introduce innovative concepts and methodologies to enhance the treatment of gastrointestinal diseases in veterinary medicine. This approach not only provides support for veterinary research and therapeutic strategies but also holds promise for addressing the challenge posed by drug-resistant strains.

## 2. Materials

### 2.1. Strains and Reagents

*E. coli* strain ATCC25922 is maintained by our laboratory. Single colonies of this strain were aseptically transferred into Luria-Bertani liquid medium and cultured on a shaker at 220 rpm for 12 h at 37 °C. Following incubation, the cultured bacteria were centrifuged at 8000 rpm for 5 min at 4 °C. The resulting supernatant was carefully discarded, and the bacterial pellet underwent three consecutive washes with sterile normal saline. Subsequently, the bacteria were resuspended in sterile normal saline, and the concentration was adjusted to 1 × 10^8^ CFU/mL.

Peptide OH-CATH30 (KFFKKLKNSVKKRAKKFFKKPRVIGVSIPF) was synthesized by the solid-phase method with a purity of 98% by Anhui Guotai Biotechnology Co., Ltd., Hefei, China. Before use, equilibrate at room temperature for 30 min, then thoroughly dissolve in normal saline.

The detailed synthesis and characterization methods for these microspheres can be found in our earlier work [19]. In summary, 30 mg of PLGA was dissolved in 5 mL of dichloromethane, and 10 mg of OH-CATH30 was dissolved in 1 mL of sterile water. The aqueous peptide solution was added dropwise to the organic polymer phase to create a primary emulsion (W_1_/O), which was intermittently sonicated for 60 s in an ice water bath at 100 W amplitude. The primary emulsion was then added by syringe to a 10 mL 0.75% (*w/v*) PVA solution containing 1% NaCl. After 30 s of sonication in an ice water bath at 60 W amplitude to form a W/O/W solution, it was added to a 15 mL 0.5% PVA solution and stirred at 400 rpm for 4 h at room temperature. The resulting emulsion underwent centrifugation (4 °C, 3000 rpm, 15 min), and the microspheres were collected, washed to remove excess PVA, lyophilized for 48 h, and stored at −20 °C until further use. The dissolution method is the same as the peptide.

Positive control drug: Gentamicin sulfate, a 15 mg/mL sterile solution purchased from Sangon Biotech (Shanghai) Co., Ltd., Shanghai, China.

### 2.2. Mice

Sixty 8-week-old male Kunming mice (25 g ± 1 g), bred under specific pathogen-free (SPF) conditions, were procured from Qingdao Darenfucheng Animal Technology Co., Ltd., Qingdao, China. They were accommodated in a specific pathogen-free animal facility at Qingdao Agricultural University, which provided ad libitum access to drinking water and commercial chow. The ambient conditions were maintained at a temperature of 24 ± 2 °C, a relative humidity of 60 ± 5%, and a 12-h light–dark cycle. A one-week acclimatization period in the laboratory preceded the commencement of experiments. All animal procedures adhered to the guidelines established by the Animal Ethics Committee of Qingdao Agricultural University (approval number: QAU2023080101).

### 2.3. Modeling of Intestinal Bacterial Infection

On the day prior to the experiment, all mice underwent a 12-h fasting period. Fifty mice were randomly selected and received a 0.2 mL solution of *E. coli* (1 × 10^8^ CFU/mL) through an 8#45 mm gastric gavage needle, repeated 6 h later. An additional 10 mice received 0.2 mL of normal saline by the same gastric gavage method, forming the healthy control group (CG). This group is designed to provide baseline data on normal physiological conditions. Following this, all mice resumed regular dietary and water intake. Behavioral changes, appetite, and other clinical symptoms were observed and recorded to evaluate potential physiological and behavioral alterations induced by the experiment. After the last gavage 12 h later, mice exhibiting noticeable symptoms following the *E. coli* gavage were randomly divided into four groups, each consisting of 10 mice: the Gentamicin sulfate-treated group (GS), the OH-CATH30-treated group (OC), the PLGA-OH-CATH30-treated group (POC), and the model group (MG). The model group did not receive any specific treatment, serving as a baseline to understand disease progression and providing a reference for comparing treatment effects.

### 2.4. Treatment Protocols and Sample Collection

The treatment for mice began 12 h after modeling and lasted for five consecutive days through gavage administration. The CG and MG groups were administered a daily gavage dose of 0.2 mL of normal saline. The GS group was treated with gentamicin sulfate (0.2 mg/day), the OC group received OH-CATH30 treatment (0.2 mg/day), and the POC group received treatment with PLGA-loaded OH-CATH30 microspheres (1.1 mg/day, with an effective peptide content of 0.2 mg).

Throughout the experimental period, observations were made regarding the mice’s feces, fur, and mental status. Furthermore, the body weights of mice in each group were measured and recorded at specified daily time points. Following the final gavage, the mice underwent a 24-h fasting period. The mice were anesthetized with ether, and blood was drawn from the orbital venous plexus. The mice were then euthanized by cervical dislocation. Spleens were isolated and weighed promptly to determine the spleen index. The isolated colon, cecum, and jejunum were cut into 1-centimeter sections and fixed in a 10% formalin solution. Fresh fecal samples were collected and stored at −80 °C. 

The variations in body weight and spleen index were calculated with Equations (1) and (2), respectively, as follows: Variations in body weight (%) = (Current Weight/Weight on Day 0) ×100%(1)
Spleen index (%) = (Spleen weight in grams/Body weight in grams) × 100%(2)

### 2.5. Fecal Microbial Count

Fecal microflora was assessed and quantified using a dilution counting method. Fresh fecal samples were suspended in sterile PBS, vortexed (0.5 g of fresh feces dissolved in 5 mL of sterile PBS), and 50 μL of fecal tissue dilution was aseptically spread onto MacConkey agar plates (Sinopharm Pharmaceutical Co., Beijing, China) [20]. The plates were then incubated for 12 h at 37 °C to enumerate colony-forming units (CFU). Results are expressed as CFU/g of feces, and the assay was performed in triplicate.

### 2.6. Blood Analysis of Mice

Mouse blood samples were carefully collected into EDTA anticoagulated tubes, gently stirred to prevent clotting, and then analyzed for mouse leukocyte and neutrophil ratios using the Abaxis VetScan HM5 (Abaxis, Inc., Union City, CA, USA). Blood was collected from mice in blood collection tubes for blood biochemistry, left to stand at room temperature for 30 min, centrifuged at 3000 rpm for 10 min, and then analyzed using an automatic biochemistry analyzer for blood biochemistry indexes manufactured by Amperex FILM (Beijing Anapure Bioscientific Co., Ltd., Beijing, China), which included the following items: Alanine transaminase (ALT), alkaline phosphatase (ALP), total bilirubin (TBIL), blood urea nitrogen (BUN), and serum creatinine (CREA).

### 2.7. Histopathological Evaluation (HE)

The mouse jejunum, cecum, and colon were carefully rinsed out with a clean syringe aspirated with normal saline until free of any impurities, fixed in a 10% formalin solution, embedded in paraffin, and stained with hematoxylin and eosin (H&E). Histological alterations were examined and analyzed using light microscopy.

### 2.8. Measurement of Cellular Inflammatory Factors

Mouse blood was collected in centrifugation tubes, left at room temperature for 2 h, and centrifuged at 3000 rpm for 10 min. The supernatant was aspirated, dispensed into new centrifugation tubes, and stored at −20 °C. When thawing, the blood was thawed at 4 °C for 20 min and then equilibrated at room temperature for 40 min. Tumor necrosis factor-α (TNF-α), interleukin-6 (IL-6), interleukin-10 (IL-10), and interleukin-1β (IL-1β) were purchased from Shanghai Enzyme-linked Biotechnology Co., Ltd., Shanghai, China, and the experiments were performed according to the instructions.

### 2.9. 16S rRNA Gene Sequencing and Analysis

Total genomic DNA was extracted from fecal samples using the TGuide S96 Magnetic Soil/Stool DNA Kit (Tiangen Biotech (Beijing) Co., Ltd., Beijing, China) according to the manufacturer’s instructions. The hypervariable region V3-V4 of the bacterial 16S rRNA gene was amplified with primer pairs 338F: 5′-ACTCCTACGGGAGGCAGCA-3′ and 806R: 5′-GGACTACHVGGGTWTCTAAT-3′. PCR products were checked on agarose gel and purified through the Omega DNA purification kit (Omega Inc., Norcross, GA, USA). The purified PCR products were collected, and the paired ends (2 × 250 bp) were performed on the Illumina Novaseq 6000 platform (Beijing Biomarker Technologies Co., Ltd., Beijing, China). The qualified sequences with more than 97% similarity thresholds were allocated to one operational taxonomic unit (OTU) using USEARCH (version 10.0). Taxonomy annotation of the OTUs/ASVs was performed based on the Naive Bayes classifier in QIIME2 (2020.6) [21] using the SILVA database [22] (release 138.1) with a confidence threshold of 70%. Alpha was performed to identify the complexity of species diversity in each sample utilizing QIIME2 software. Beta diversity calculations were analyzed by principal coordinate analysis (PCoA) to assess the diversity in samples for species complexity. A one-way analysis of variance was used to compare bacterial abundance and diversity. Linear discriminant analysis (LDA) coupled with effect size (LEfSe) [23] was applied to evaluate the differentially abundant taxa. The online platform BMKCloud (https://www.biocloud.net (accessed on 12 October 2023)) was used to analyze the sequencing data. The 16S rRNA sequencing data presented in the study are deposited in the NCBI repository, accession number PRJNA1035650.

### 2.10. Statistical Analyses

The data were analyzed using GraphPad Prism 8.0.2 statistical software, and the experimental results are presented as means ± standard deviation (SD). Statistical analysis involved one-way analysis of variance (ANOVA) for comparisons among multiple groups, unless otherwise stated. Significance levels were denoted as ** p* < 0.05, *** p* < 0.01, **** p* < 0.001, ***** p* < 0.0001, and “ns” indicates non-significant. *p <* 0.05 was regarded as statistically significant.

## 3. Results

### 3.1. Clinical Symptoms

During the experiment, six of the 50 mice gavaged with *E. coli* died, while the rest developed observable symptoms such as diarrhea, lethargy, increased aggregation behavior, and fur loss. Subsequently, a cohort of 40 mice was systematically selected from the surviving population and randomly allocated into four distinct treatment groups. Body weight serves as a vital metric for assessing the mice’s overall health status. In Figure 1A, during the initial 1–2 days of treatment, all groups significantly decreased in body weight compared to the CG group (*p* < 0.01). From the third day on, body weight generally increased across all groups. Particularly, the POC and GS groups exhibited more pronounced weight gain than the OC and MG groups. Throughout the 5-day treatment, the weight trajectories of the POC and GS groups showed a gradual increase, approaching the levels observed in the CG group. The weight increase trend in the GS group was weaker than in the POC group, possibly due to GS-induced appetite suppression. In Figure 1B, we evaluated the splenic index, a specific physiological parameter associated with splenic function. A noteworthy increase in the splenic index was discerned in mice within the MG and OC groups when contrasted with the CG group. However, no statistically significant variance in the splenic index was observed between the POC and GS groups and the CG group. Figure 1C illustrates the *E. coli* bacterial load within the fecal samples. In comparison to the CG group, there was a substantial elevation in the *E. coli* count within the MG and OC groups, whereas a marked reduction in the *E. coli* population was evident in the POC and GS groups. This phenomenon can be attributed to the effective clearance of *E. coli* strains within the host’s intestinal environment following POC and GS treatments.

### 3.2. Blood Analysis

Intestinal bacterial infections have a profound impact on the integrity of the intestinal mucosal barrier, resulting in an increase in intestinal pathogenic bacteria, the release of endotoxins, systemic inflammatory responses, and organ damage, as noted by Wang et al. [24]. To comprehensively assess the immune response of antimicrobial peptide microspheres to infection and organ damage in mice, we conducted an analysis of routine blood and blood biochemical parameters. Figure 2A depicts an elevated leukocyte count and an increased percentage of neutrophils in the MG and OC groups in comparison to the CG group. However, in the POC and GS groups, these two indicators were normalized, indicating an improvement in the inflammatory response in the body and highlighting the positive impact of the treatment. Figure 2B demonstrates that the five blood biochemical indices measured in the POC and GS groups remained largely unaffected compared to those in the CG group. However, distinct differences emerged in the liver-related indices, specifically alkaline phosphatase (ALP) and total bilirubin (TBIL), as well as the kidney-related indices, creatinine (CREA), and urea nitrogen (BUN), when comparing the MG group with the CG group. These results underscored the potential adverse effects of untreated *E. coli* infections on liver and kidney functions in mice. Furthermore, these findings substantiate the efficacy of the prepared microspheres in preserving the liver and kidney functions of bacterially infected mice. Importantly, these microspheres demonstrate a favorable safety profile in the context of drug administration, as they do not induce hepatotoxicity or nephrotoxicity.

### 3.3. Effect of MS on Inflammation and Intestinal Damage in Mice with Bacterial Enteritis

In a previous study [25], it was established that serum cytokines can serve as indicators of mucosal inflammation. In our research, we used an enzyme-linked immunosorbent assay (ELISA) to measure serum concentrations of IL-10, IL-6, TNF-α, and IL-1β in mice to assess the impact of microsphere treatment on *E. coli*-induced intestinal inflammation. Figure 3A illustrates a significant reduction in IL-10 levels in the MG and OC groups compared to the CG group. Post-treatment in the POC and GS groups normalized IL-10 levels. Pro-inflammatory factors IL-6, TNF-α, and IL-1β significantly increased in the MG and OC groups compared to the CG group. However, POC and GS treatments led to a marked decrease in these factors. These findings highlight the crucial role of microsphere therapy in mitigating inflammation and preserving immune balance. Regarding histopathological findings (Figure 3B), the mucosal epithelium of the jejunum, cecum, and colon in the CG group displayed structural integrity, orderly arrangement, and regular villous architecture. In contrast, in the MG group, the colonic mucosal epithelium exhibited incompleteness, characterized by inflammatory cell infiltration within the lamina propria, along with partial cell necrosis and detachment in the mucosal layer. The jejunum exhibited structural disruption of the villi, detachment, substantial inflammatory cell infiltration, and some vacuolization. The cecum presented mucosal atrophy and edema, along with crypt hyperplasia. Both GS and POC treatments led to a noticeable improvement in intestinal tissues, with reduced cellular necrosis and detachment and the preservation of a relatively intact mucosal layer, along with a decrease in vacuolization. In contrast, the OC treatment group did not exhibit significant symptom improvement. These findings emphasize the beneficial role of the prepared microspheres in mitigating *E. coli*-induced intestinal damage while indicating that the direct application of antimicrobial peptides yields less favorable results.

### 3.4. Effects of MS on Gut Microbiota Diversity and Community Composition

In individuals with enteritis, there was a significant disruption in the composition and function of their gut microbiota, supported by robust empirical evidence [26]. To comprehensively investigate the impact of microspheres on the intestinal microbiota, we conducted 16S rRNA sequencing analysis of fecal samples from five distinct groups of mice. We employed relative abundance curves, a crucial tool for elucidating species abundance and community evenness. Along the horizontal axis, the curve width reflects species abundance, with wider curves indicating greater species abundance. Curve smoothness signifies the uniformity of species distribution within the samples, with smoother curves indicating a more even distribution of species within the community. Figure 4A illustrates that the curves for the POC and GS groups are relatively narrower compared to the control group, implying lower species abundance. However, as sequencing depth increased, the data from all five groups exhibited a trend towards smoother curves, underscoring the sufficiency of sequencing depth in capturing species diversity within the samples. Alpha diversity serves as a fundamental metric for assessing species richness and diversity within ecosystems [27]. Among them, the Chao and Ace indices quantified variations in bacterial species abundance among groups in the gut microbiota, while the Shannon and Simpson indices assessed alterations in bacterial species diversity within the gut microbiota [28]. As depicted in Figure 4B, our analysis of the Chao and Ace indices, along with the Shannon and Simpson indices, consistently indicated a substantial decrease in microbial diversity and species richness in the POC-treated and GS-treated groups when compared to the control group. Notably, the GS group exhibited more pronounced changes. In contrast, both the model group and the OC treatment group showed only a minor decrease.

In our analytical approach, we relied on OTU data and employed the binary Jaccard distance algorithm. As illustrated in Figure 4C, our Principal Coordinate Analysis (PCoA) revealed significant variations in the composition of the gut microbiota among the five groups of mice (*p* < 0.005). To gain a comprehensive understanding of the impact of different treatment regimens on the intestinal bacterial community of mice, we generated bacterial distribution bar charts, visually representing the relative abundance distribution of bacterial species in each sample. This visualization facilitates the comparison of distinctions among different samples. According to the findings presented in Figure 4D, at the phylum level, we observed that within each group, the two pivotal bacterial groups, Firmicutes and Bacteroidetes, represented a substantial proportion of the mouse intestinal population. However, both the OC and MG groups demonstrated a reduction in the abundance of *Bacteroidetes* and *Firmicutes* compared to the control group while showing an increase in the relative abundance of *Proteobacteria*. Notably, POC treatment induced structural alterations at the phylum level within the mouse intestinal microbiota, characterized by an increased abundance of *Bacteroidetes* and a reduced abundance of *Proteobacteria*. Particularly noteworthy, the GS-treated group experienced a significant decline in *Firmicutes* abundance and a considerable increase in *Bacteroidota* abundance compared to the POC-treated group. At the genus level, observations indicated that in the model group, the relative abundance of *Escherichia Shigella* and *Parasutterella* significantly exceeded that of the control group. However, post-POC treatment, there was an observable declining trend in the relative abundance of these two genera. Concurrently, following POC treatment, there was an increase in the relative abundance of *Blautia* and Lactobacillus. In contrast, the OC treatment group exhibited a pattern closely resembling that of the model group, while the GS treatment group experienced a substantial reduction in the relative abundance of several genera, with certain genera nearly disappearing entirely.

Subsequently, LEfSe analysis was used to screen out the enrichment of vital phylotypes according to taxon analysis. Figure 4E presents the essential biomarkers for each group, visually displayed in both the cladogram and the histogram depicting the LDA score (LDA ≥ 4). The model group exhibited significantly higher levels of *Gammaproteobacteria* and *Burkholderiales*, among others. In contrast, the POC group displayed a simpler composition, with the prevalence of species such as *Phocaeicola vulgatus* and *Bacteroides caecimuris.*

## 4. Discussion

Antimicrobial peptides (AMPs) are small peptides found widely in nature, serving as integral components of the innate immune system in various organisms [29]. Renowned for their low toxicity, thermal stability, and broad-spectrum antimicrobial properties, AMPs have emerged as promising alternatives to traditional antimicrobial drugs [30]. However, realizing their potential in treating intestinal bacterial infections has been challenging due to the complexity of the organism’s intestinal environment. Presently, AMPs are primarily administered through injections, posing a potential barrier to patient compliance, especially in cases requiring long-term treatment with multiple injections [31]. Moreover, oral administration of antimicrobial peptides has encountered challenges in achieving notable bactericidal or therapeutic effects [32,33]. This limitation primarily results from the complex gastrointestinal environment, including factors like gastric acid and digestive enzymes, limiting effective AMP concentrations at the site of inflammation [34]. Building on our previous studies, we provide evidence indicating that the PLGA-loaded antimicrobial peptide OH-CATH30 microspheres display notable bactericidal efficacy both in vitro and on the ocular surface. Additionally, OH-CATH30, a member of the *cathelicidin* family, consistently demonstrated effectiveness against a diverse range of pathogens, establishing itself as a promising candidate for innovative therapeutic applications [35]. This evidence led us to explore the technology’s application in addressing intestinal infections. In the current study, microsphere treatment successfully mitigated adverse clinical manifestations associated with intestinal *E. coli* infections in mice, providing significant benefits such as reduced inflammation and improved intestinal health. Importantly, the efficacy of the microspheres was on par with that of the conventional antibiotic gentamicin sulfate, highlighting their competitiveness in terms of therapeutic efficacy. Furthermore, due to the inherent biodegradability of PLGA and its relatively minor impact on experimental animals [36,37], we excluded the inclusion of a control group using empty PLGA microspheres in the experimental design. Additionally, anticipating potential intricacies and interpretive challenges that could introduce physiological variations, no drug control was implemented in the healthy group, intentionally aiming to facilitate a focused evaluation of the drug’s effects specifically in the infected state and enhance the clarity and interpretability of the study results.

Research indicates that disruptions in the integrity of the intestinal mucosal barrier lead to increased intestinal permeability, allowing the unrestricted passage of luminal contents and microorganisms through the intestinal lamina propria [38,39]. In such scenarios, the organism depends on immune responses and cellular repair mechanisms to eliminate potentially harmful substances, thereby maintaining overall gastrointestinal well-being [40,41]. Nonetheless, in the MG and OC groups, our observations unveiled a deficiency in these interventions, leading to a prolonged recovery process of the compromised intestinal barrier. This coincided with an increase in TNF-α, an inflammatory mediator recognized for its ability to stimulate the secretion of other pro-inflammatory cytokines, such as IL-1β and IL-6. This, in turn, enhances the adhesion and infiltration of leukocytes into the inflamed and damaged intestinal mucosal tissues [42,43,44]. This cascade of inflammatory events may adversely impact the health of the liver and kidneys, as evidenced by modifications in liver and kidney-related parameters in contrast to the CG group. However, efficacious treatment with POC and GS substantially ameliorated the damage to the intestinal barrier and mitigated unfavorable physiological alterations in enteritis-afflicted mice. Another important finding was that in the POC and GS groups, the heightened levels of the anti-inflammatory cytokine IL-10 signify the transition of enteritis towards a recovery phase, and IL-10 plays a critical role in maintaining the equilibrium between pro-inflammatory and anti-inflammatory factors, averting an undue inflammatory response [45,46]. In general, microsphere therapy may be effective in treating enteritis by lowering the levels of pro-inflammatory cytokines, increasing the levels of anti-inflammatory cytokines, and promoting a balance between the dynamics of these two factors.

The intestinal microbiota plays a crucial role in the host’s metabolic processes and the development and functioning of the immune system [47]. A substantial body of research has demonstrated that disruptions in the equilibrium of the gut microbiota can lead to immune system dysregulation and the excessive proliferation of potentially pathogenic microorganisms, including pathogenic bacteria, ultimately resulting in a range of inflammatory bowel diseases, such as Crohn’s disease and ulcerative colitis [48,49]. According to the findings by Stecher et al., a significant reduction in microbial diversity is a common observation in the majority of patients with enteritis. However, our present study revealed that the decrease in microbial diversity within the intestinal tract of mice in the model group was not significant [50]. This variation may be attributed to a range of factors, including host-specific characteristics, the choice of infection models, the specific treatment regimens, and various environmental variables, all of which contribute to variations in research outcomes. This underscores the intricate interplay between enteritis and the composition of the gut microbiota.

In the model group, despite the absence of a significant decrease in species diversity, an increase in the number of harmful bacterial groups led to a disturbance in the core microbiota, resulting in the emergence of pronounced clinical symptoms. At the phylum level, the model group exhibited a reduction in the relative abundance of *Bacteroidetes* and *Firmicutes* compared to the control group, accompanied by an elevation in the relative abundance of *Proteobacteria*, which is consistent with the findings reported by Organ XC et al. [51]. Furthermore, the research conducted by Wang M et al. [52] demonstrated that *Proteobacteria* might contribute to intestinal inflammation through the production of lipopolysaccharide (LPS) and immunostimulatory flagellin, in conjunction with their direct interactions with the host’s immune responses. Additionally, at the genus level, the model group displayed an increased abundance of *Escherichia Shigella* and *Parasutterella*. *Escherichia Shigella*, characterized as an amoeboid bacterium, incites intestinal inflammation by breaching epithelial cells, leading to macrophage apoptosis and the release of IL-1β [53]. *Parasutterella* has been identified as a bacterium that predisposes individuals to enteritis and sepsis [54]. 

However, following POC treatment, harmful bacteria decreased, while beneficial bacteria such as *Phocaeicola vulgatus* and *Blautia* experienced an increase. Notably, *Phocaeicola vulgatus* is recognized for its beneficial attributes, including the regulation of intestinal pH, inhibition of the growth of pathogenic bacteria, and facilitation of the colonization of beneficial bacteria [55,56]. Similarly, *Blautia*, a newly discovered potential probiotic in recent years, plays a pivotal role in maintaining the equilibrium of the host’s intestinal environment and preventing inflammation [57]. However, the OC treatment group exhibited analogous alterations in gut flora when compared to the model group, providing further confirmation that the direct oral administration of antimicrobial peptides did not significantly ameliorate symptoms. Conversely, the GS-treated group displayed changes in intestinal flora similar to those of the POC group, but the impact of GS treatment on the intestinal flora was more conspicuous. These studies offer additional validation for the effective use of the antimicrobial peptide OH-CATH30, when encapsulated in PLGA and administered orally, in mitigating *E. coli*-induced intestinal dysfunction by preserving intestinal homeostasis and enhancing the populations of beneficial intestinal bacteria. This underscores OH-CATH30’s capacity to modulate the potential of the gut microbiota following successful in vivo administration, thereby opening up new possibilities for the development of more effective treatment strategies and drug delivery systems. However, to elucidate the specific mechanisms by which microspheres influence changes in the intestinal microbiota of mice, further research is needed.

## 5. Conclusions

In summary, our study emphasizes the significant efficacy of PLGA-OH-CATH30 microspheres in countering *E. coli* proliferation, inflammation, and maintaining intestinal integrity. The positive impact on murine immune and inflammatory responses not only helps maintain intestinal homeostasis but also promotes beneficial bacterial growth. These findings provide a basis for exploring therapeutic applications of PLGA-OH-CATH30 microspheres, particularly in addressing inflammatory bowel diseases. Despite the focus on a mouse model, there are promising implications for treating inflammatory bowel diseases and exploring oral antimicrobial peptide applications. Building upon the success of microsphere therapy, our future research will be focused on seeking more effective antimicrobial peptides and integrating them into microsphere technology. Through a comprehensive exploration of molecular mechanisms, our objective is to refine the antimicrobial therapeutic strategy, enhance therapeutic efficacy, and potentially offer innovative and effective pathways for the development of novel veterinary drugs in clinical practice.

## Figures and Tables

**Figure 1 biomolecules-14-00086-f001:**
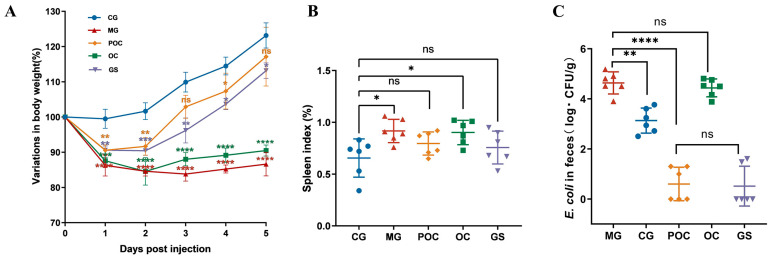
(**A**) Variation in body weights (%) of mice in each group. The significance of intergroup differences was evaluated using a two-way ANOVA with a Tukey post hoc test. (**B**) Splenic index of mice in each group. (**C**) Number of bacteria in the feces of each group of mice. Values are reported as mean ± SD, *n* = 6. ** p* < 0.05, *** p* < 0.01, **** p* < 0.001, ***** p <* 0.0001, and “ns” indicates non-significant.

**Figure 2 biomolecules-14-00086-f002:**
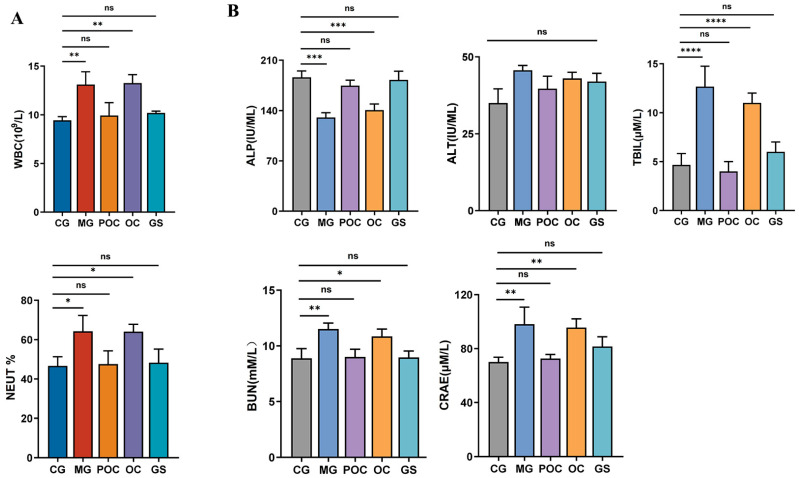
(**A**) Mouse blood routine: white blood cell count and percentage of neutrophils in blood. (**B**) Correlation indices for nephrotoxic and hepatotoxic injuries. Values are reported as mean ± SD, *n* = 3. ** p* < 0.05, *** p* < 0.01, **** p* < 0.001, ***** p* < 0.0001, and “ns” indicates non-significant.

**Figure 3 biomolecules-14-00086-f003:**
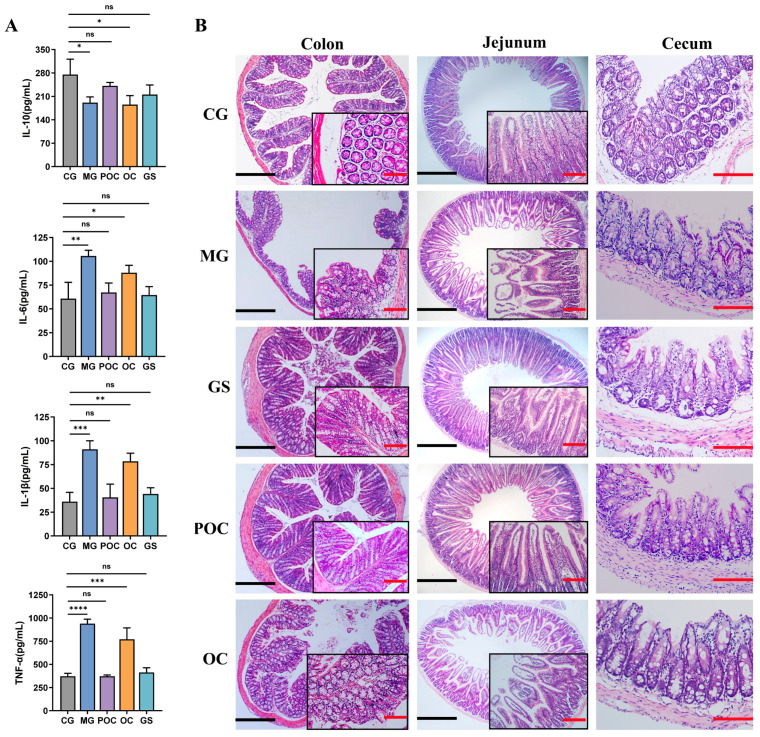
(**A**) Effect of serum levels of IL10, IL-6, TNF-α, and IL-1β in different groups of mice. Values are reported as mean ± SD, *n* = 3. * *p* < 0.05, ** *p* < 0.01, *** *p* < 0.001, **** *p* < 0.0001, and “ns” indicates non-significant. (**B**) The H&E staining of the mouse colon, jejunum, and cecum. The scale bars are used to indicate size; the black scale represents 500 μm, and the red scale represents 50 μm.

**Figure 4 biomolecules-14-00086-f004:**
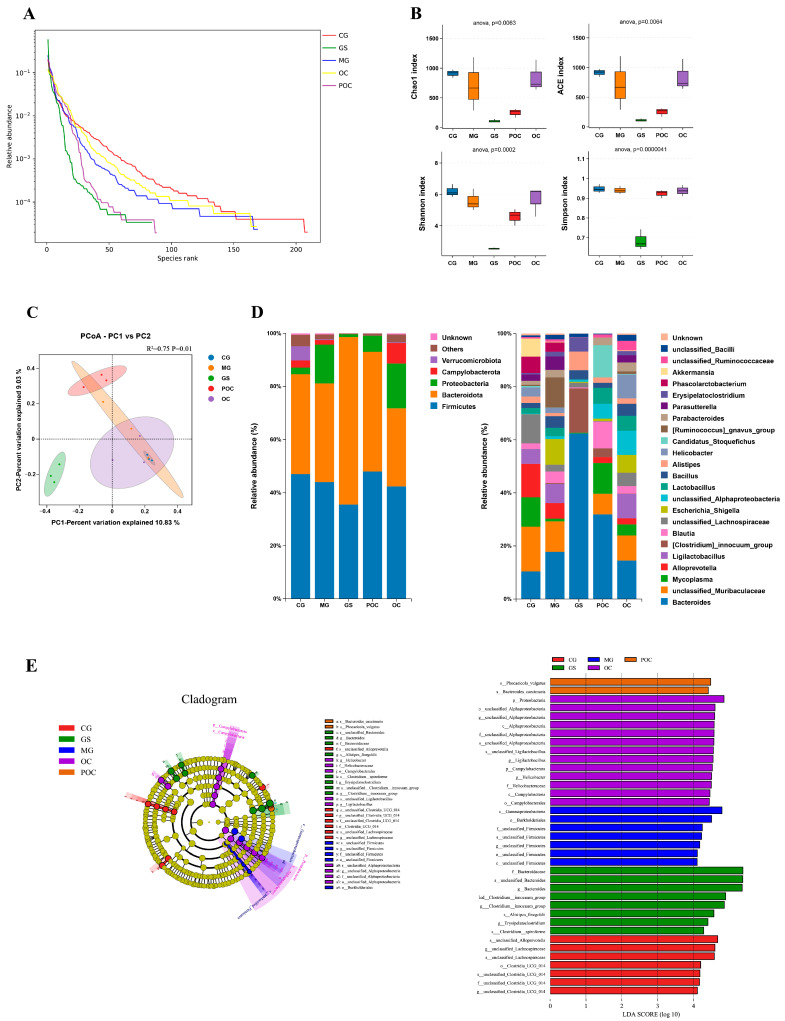
(**A**) Rank abundance curves assessing sample richness and evenness in the feces of mice. (**B**) α-diversity indexes of gut microbiota among five groups. (**C**) PCoA based on OTU level. (**D**) Microbial community composition at the phylum and genus levels. (**E**) Taxonomic cladogram obtained by LEfSe and LDA score distribution histogram (LDA  ≥  4.0).

## Data Availability

The authors declare that all additional data supporting the study’s findings can be found within the paper or may be obtained by contacting the corresponding author(s) upon request.

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
