# Peer review of "Exploring PLGA-OH-CATH30 Microspheres for Oral Therapy of Escherichia coli-Induced Enteritis"

_biomolecules, 2024, doi:10.3390/biom14010086_

Round 1
Reviewer 1 Report
Comments and Suggestions for Authors
This paper describes the effects of microspheres coated with an antimicrobial peptide on mice infected with E. coli. I find the work interesting, but I have some minor concerns:
Why did the authors describe the amount of peptide administered to the mice as concentration? Actually, I need help understanding how the microspheres were administered. How many milligrams, or micrograms, were administered to each mouse?
How many mice were in total? Six mice died of how many?
Even though the microspheres with the AMP were previously reported, please include their characterization here.
What about other AMPs? Could this technology be applied to more AMPs with lower MICs than OH-CATH30?
Escherichia coli and E. coli are not italicized in the manuscript and references.
Comments on the Quality of English LanguageI suggest to rewrite the abstract and have an English expert proofread it.
The materials and methods section must be written more clearly for better understanding.
Reviewer 2 Report
Comments and Suggestions for Authors
The manuscript entitled “Exploring PLGA-OH-CATH30 Microspheres for Oral Therapy of Escherichia coli-Induced Enteritis” presented the effect of PLGA-loaded antimicrobial peptide OH-CATH30 microspheres for the treatment of enteritis in mice. The manuscript is scientifically sound but requires minor revision before acceptance for publication.
1. In the introduction section; give the background of the OH-CATH30 peptide. How does it develop? What are its characteristics? What is its activity?
2. In the experiment section; give the animal permission number in line 109.
3. Line 125; explain the model group (MG) used in this study.
4. Fig 2 and 3; add the information that the significant data was compared to the CG group.
5. Line 245; give the full name of MS.
6. In the discussion section; What is the stability of OH-CATH30 peptide in the microspheres? In addition, how did the peptide microspheres support the increased abundance of beneficial bacteria?
Comments on the Quality of English LanguageMinor editing of English language required.
Reviewer 3 Report
Comments and Suggestions for Authors
The manuscript is very well done and generally the conclusions are supported by the data. The experiments are generally well designed, with one exception (below).
1. In figure 1A, was there any statistical difference between the groups regarding body weight? If not, perhaps the authors can show some correlation graph between body weight and spleen index (only available post mortem) showing that these actually do correlate.
2. The major concern experimentally is that there is not (A) PGLA-only control experiments and (B) peptide-only controls on UNinfected mice. It is also unclear what the difference is between the CG and MG. (1) Without an "empty" pgla control, there is difficulty in definitively determining the meaning of the results. Is there any literature information on how PGLA spheres affect these factors? (2) without peptide-only control on UNinfected mice, we cannot determine what variability on microflora is "expected" from the peptide treatment and how that varies with the addition of the infection.
3. The authors should describe and provide references to the indices used in Figure 4B. References are required, but a short sentence or two describing WHAT these different indices look for is appropriate too. Also, i believe one of the graphs in 4B is mislabeled, as two of them are Chao and none are labeled simpson.
4. Overall the manuscript could use some editing. There are examples of awkward wording (e.g. erosion on line 349), There are also numerous instances of gratuitous adjectives/descriptors in the text. While the final decision will be the editor's on this point, i feel that the strong, convincing data stands on it's own.
Comments on the Quality of English Language
Generally fine except for overly flowery language.
Round 2
Reviewer 3 Report
Comments and Suggestions for Authors
The rationale behind the choices in control groups for the animal studies are well explained in the response letter. I must insist that these be included, in a more abbreviated form, in the manuscript itself. Just 2-3 sentences in the discussion should be sufficient, along with the references provided in the response letter.
Comments on the Quality of English Languagefine
